# Functional Role of Fatty Acid Synthase for Signal Transduction in Core-Binding Factor Acute Myeloid Leukemia with an Activating c-Kit Mutation

**DOI:** 10.3390/biomedicines13030619

**Published:** 2025-03-03

**Authors:** Ruimeng Zhuang, Bente Siebels, Konstantin Hoffer, Anna Worthmann, Stefan Horn, Nikolas Christian Cornelius von Bubnoff, Cyrus Khandanpour, Niklas Gebauer, Sivahari Prasad Gorantla, Hanna Voss, Hartmut Schlüter, Malte Kriegs, Walter Fiedler, Carsten Bokemeyer, Manfred Jücker, Maxim Kebenko

**Affiliations:** 1Institute of Biochemistry and Signal Transduction, Center for Experimental Medicine, University Medical Center Hamburg-Eppendorf, 20246 Hamburg, Germany; ruimengzh@163.com (R.Z.); juecker@uke.de (M.J.); 2Section Mass Spectrometric Proteomics, University Medical Center Hamburg-Eppendorf, 20246 Hamburg, Germany; bente.siebels@googlemail.com (B.S.); hschluet@uke.de (H.S.); 3Department of Radiobiology & Radiation Oncology, UCCH Kinomics Core Facility, University Cancer Center Hamburg, University Medical Center Hamburg-Eppendorf, 20246 Hamburg, Germany; k.hoffer@uke.de (K.H.); m.kriegs@uke.de (M.K.); 4Department of Biochemistry and Molecular Cell Biology, University Medical Center Hamburg-Eppendorf, 20246 Hamburg, Germany; a.worthmann@uke.de; 5Research Department Cell and Gene Therapy, Department of Stem Cell Transplantation, University Medical Center Hamburg-Eppendorf, 20246 Hamburg, Germany; shorn@uke.de; 6Clinic for Hematology and Oncology, University Hospital Schleswig-Holstein Campus, 23562 Lübeck, Germany; nikolaschristiancornelius.vonbubnoff@uksh.de (N.C.C.v.B.); cyrus.khandanpour@uksh.de (C.K.); niklas.gebauer@uksh.de (N.G.); sivahari.prasadgorantla@uksh.de (S.P.G.); 7Hubertus Wald Tumorzentrum, Department of Oncology–Hematology, Bone Marrow Transplantation and Pneumology, University Cancer Center, 20251 Hamburg, Germany; fiedler@uke.de (W.F.); c.bokemeyer@uke.de (C.B.)

**Keywords:** fatty acid synthase, palmitoylation, TVB-3166, AML, c-Kit mutation, PI3K, Akt, mTOR, S6 kinase, Lyn, Gli1, hedgehog signaling

## Abstract

**Background/Objectives**: Acute myeloid leukemia (AML) is a rare hematological malignancy with a poor prognosis. Activating c-Kit (CD117) mutations occur in 5% of de novo AML and 30% of core-binding factor (CBF) AML, leading to worse clinical outcomes. Posttranslational modifications, particularly with myristic and palmitic acid, are crucial for various cellular processes, including membrane organization, signal transduction, and apoptosis regulation. However, most research has focused on solid tumors, with limited understanding of these mechanisms in AML. Fatty acid synthase (FASN), a key palmitoyl-acyltransferase, regulates the subcellular localization, trafficking, and degradation of target proteins, such as H-Ras, N-Ras, and FLT3-ITDmut receptors in AML. **Methods**: In this study, we investigated the role of FASN in two c-Kit-N822K-mutated AML cell lines using FASN knockdown via shRNA and the FASN inhibitor TVB-3166. Functional implications, including cell proliferation, were assessed through Western blotting, mass spectrometry, and PamGene. **Results**: FASN inhibition led to an increased phosphorylation of c-Kit (p-c-Kit), Lyn kinase (pLyn), MAP kinase (pMAPK), and S6 kinase (pS6). Furthermore, we observed sustained high expression of Gli1 in Kasumi1 cells following FASN inhibition, which is well known to be mediated by the upregulation of pS6. **Conclusions**: The combination of TVB-3166 and the Gli inhibitor GANT61 resulted in a significant reduction in the survival of Kasumi1 cells.

## 1. Introduction

AML is a heterogeneous disease characterized by a block in differentiation and uncontrolled proliferation of myeloid blasts [1,2]. Currently, diverse recurrent chromosomal and molecular aberrations with prognostic and predictive impact are known and increasingly used in clinical practice [3,4].

Chromosomal aberrations such as t(8;21)(q22;q22) and inv(16)(p13;q22), present in 12–15% of AML cases, lead to abnormal function of the transcriptional CBF complex [5]. While activating c-Kit mutations are found in only 5% of patients with de novo AML, their prevalence rises to approximately 30% in the CBF-AML subgroup [6]. These mutations primarily affect Exon 8 (extracellular domain) or Exon 17 (internal tyrosine kinase domain), and are associated with adverse clinical outcomes [7]. Ongoing trials are investigating the use of c-Kit inhibitors in combination with standard chemotherapy [8]. CD117/c-Kit was initially called stem cell factor receptor because of its function in hematopoietic stem cell survival, self-renewal, and differentiation. It belongs to the receptor tyrosine kinase (RTK) family type III consisting of an extracellular domain, a transmembrane domain, a juxtamembrane domain (JM domaine), and an intracellular tyrosine kinase domain. Most human AML cells express wild-type c-Kit, which is constitutively autophosphorylated by binding of the ligand stem cell factor (SCF) [9]. In the case of activating mutations, a ligand-independent activation of the downstream c-Kit signaling takes place [10,11]. Various co-effector proteins such as Sos, GrB2, phosphatidylinositol-3 (PI3)-kinase, JAK2 or Src family kinases (SFK) bind to the JM domaine and C-terminal tail of c-Kit, leading to an activation of the main downstream transduction elements Ras-Raf-MAPK, JAK/STAT, and the PI3K/Akt pathway. Numerous authors reported several crosstalk mechanisms between these pathways [10,11,12]. In particular, the interaction between PI3K/AKT and Ras/Raf can synergistically mediate the phosphorylation of the RPS6KB1 (also called p70S6K)–RPS6 axis, resulting in an upregulation of ribosomal biosynthesis and cell growth [13,14,15].

Post-translational protein modification by the covalent attachment of lipids (lipidation) plays an important role for membrane association and the activity of various oncogenic drivers such as overexpressed or mutant epidermal growth factor receptor (EGFR), Ras, or FLT3 receptors. The irreversible prenylation of Ras by farnesyl transferase (FTase) or geranylgeranyl transferase (GGTase) represents the most crucial mechanisms resulting in the integration of the small GTPases into the plasma membrane (PM) [16]. However, the effectiveness of FTase and GGTase inhibitors is currently limited and combined treatments are too toxic for clinical use [17]. S-palmitoylation of cysteine sites, preferentially located to the C-termini of target proteins, represents another reversible mechanism. It is required for the localization of RTKs, as well as non-RTKs such as Src kinases, on the plasma and the cytoplasmatic membranes, as well as for the subsequent activation of cell signaling [18]. Several palmitoyl acyltransferases, for example, FASN or zinc finger DHHC-type palmitoyl acyltransferases (ZDHHCs), which are mostly localized on endoplasmic reticulum (ER), can mediate palmitoylation. FASN activity can be regulated in response to cell metabolism and growth signals driven by SREBP-1, ZBTB7A and p53 downstream of PI3K–Akt–mTOR, as well as MAPK signaling [19,20,21].

In this study, we investigated the role of FASN in c-Kit signaling in human and murine AML cell lines with activating c-Kit mutations. Inhibition of FASN, using either the compound TVB-3166 or shRNA, resulted in a significant increase in the pLyn-pS6 pathway, which, in turn, sustained high Gli1 expression. Consistent with these findings, combining TVB-3166 with the Gli1/2 inhibitor GANT61 significantly reduced the survival of Kasumi1 cells.

## 2. Materials and Methods

### 2.1. Materials and Reagents

#### 2.1.1. Bacterial Strain

For the bacterial strain, Xl1 blue from Pierce Thermo Fisher Scientific (Waltham, MA, USA) was used.

#### 2.1.2. Kits

DC-Protein Assay from Bio-Rad (Munich, Germany), NucleoBond Xtra Midi from Macherey-Nagel (Düren, Germany), Lipofectamine P3000 transfection reagents from Gibco/Thermo Fisher Scientific (Waltham, MA, USA), and Super Signal West Dura chemiluminescence substrate from Thermo Fisher Scientific (Waltham, MA, USA) were used.

#### 2.1.3. Antibodies

The monoclonal antibodies directed against panAkt (#4685S), pAkt S473 (#4060S), S6 (#2217S), pS6 (#2215S), MAPK (#4695S), pMAPK (#4377S), and c-Kit (Ab81) (#3308) were purchased from Cell Signaling Technology (Beverly, MA, USA), and monoclonal antibodies directed against FASN (#48357) and Gli1 (#515751) were purchased from Santa Cruz Biotechnology (Heidelberg, Germany). The anti-mouse IgG HRP-linked antibody (#7076) and anti-rabbit IgG HRP-linked antibody (#7074) were from Cell Signaling Technology (Beverly, MA, USA).

#### 2.1.4. Vectors

PLKO.1-puro vectors encoding FASN shRNA or non-target (scrambled, scr) shRNA were purchased from Sigma-Aldrich (Taufkirchen, Germany). The third-generation lentiviral vector LeGO-iB2Zeo was used to express mutant KIT N822K in conjunction with a fusion of mTagBFP and Zeocin resistance (Sh ble) in Baf3 cells. The tyrosine kinase domain mutant N822K of human Kit was generated by in vitro mutagenesis using the QuikChange Lightning Site-Directed Mutagenesis Kit (Agilent, Santa Clara, CA, USA) on wild type *KIT* cDNA [22] and N822K mutagenesis primers sthp289fw (5′-CTAGCCAGAGACATCAAGAATGATTCTAAGTATGTGGTTAAAGGAA-3′) and sthp290rv (5′-TTCCTTTAACCACATACTTAGAATCATTCTTGATGTCTCTGGCTAG-3′). The sequence encoding KIT N822K was introduced into the LeGO vector through NotI cloning and verified by sequence analysis.

#### 2.1.5. Inhibitors

TVB-3166 is owned by Sagimet Biosciences Inc. (formerly 3-V-Biosciences) (San Mateo, CA, USA) and was kindly provided for this work.

### 2.2. Methods

#### 2.2.1. Culturing of Cells

Thawing and freezing were performed according to the local directions. Cell density was maintained between 3 × 10^5^ and 3 × 10^6^ viable cells/mL. For evaluating cell concentrations, the standard trypan blue exclusion assay using a Neubauer chamber was performed. Kasumi1, a human AML cell line bearing translocation t(8;21) and Kit N822K gain-of-function mutation, and Baf3 with Kit N822K were grown in RPMI1640, supplemented with 20% FCS and 1% penicillin/streptomycin (P/S). For Baf3 wild-type cell recombinant mouse interleukin-3 was added to a final concentration of 0.5 ng/mL. HEK-293T cells were grown in DMEM supplemented with 10% FCS without P/S. Cells were protected from contamination by working under a safety cabinet class ll and cultivated in an incubator at 37 °C with 5% CO_2_.

#### 2.2.2. Proliferation

For inhibitor treatment, 10,000 Kasumi1 cells were plated in 100 μL per well in a 96-well plate (Greiner Bio-One, Frickenhausen, Germany). After 24 h, 100 μL of inhibitor solution was added. To compare the proliferation of Kasumi1 cells with and without FASN knockdown, Kasumi1 scr- and FASN-knockdown cells were plated accordingly in a 200 µL medium containing 1.5 μg/mL puromycin for different incubation times as annotated in the figures. Cell confluence was measured by the IncuCyte Zoom imaging system (Essen Bioscience, Ann Arbor, MI, USA).

#### 2.2.3. Cell Viability Assay

Kasumi1 cells with scr or FASN knockdown were counted by the standard trypan blue exclusion assay using a Neubauer chamber. Not-transfected Kasumi1 cells were incubated with the following compounds for 5d: DMSO (control), TVB-3166 at a dose range 10–250 nM or GANT61 5–50 μM as well as with individual combinations. After that, they were counted by the Vi-CELL™ XR (Beckman Coulter, Krefeld, Germany).

#### 2.2.4. Transformation and Plasmid Preparation

For the transformation and plasmid preparation, 100 μL Xl1 blue bacteria were incubated with 100 ng plasmid DNA following the manufacturer’s instruction. The NucleoBond Xtra plasmid purification Kit (Sigma-Aldrich, Taufkirchen, Germany) was used to purify plasmid DNA (see Appendix A).

#### 2.2.5. Lentiviral Knockdown of FASN

PLKO.1-puro vector encoding FASN-shRNA and PLKO.1 non-target (scrambled, scr) were purchased from Sigma-Aldrich (Taufkirchen, Germany). Two FASN-knockdown clones (kd1 and kd2) were established to check the reproducibility of the results. For virus production, HEK293T cells were plated in a DMEM medium in one 10 cm dish with 2 × 10^5^ cells per dish. For the transfection of the viral vectors and helper plasmids, the lipofectamine kit was used following the manufacturer’s instructions. In brief, 2.5 μg DNA of each PLKO.1 vector were diluted in 20 μL P3000 reagent, followed by the addition of 8 μg VSVG-, gagPol-, and HIV1-Rev-encoding plasmids. Target cells were seeded at a density of 3 × 10^5^ cells per well in a 2 mL RPMI medium. The viral supernatants were harvested 24 h and 48 h after transfection, and added immediately to the target cells. The selection of transduced target cells was carried out with 4 μg/mL puromycin and considered completed when all cells in the untransduced control wells were dead. All work with lentiviral particles was carried out in a S2 facility after approval according to the German law.

#### 2.2.6. Immunoblotting

Protein extracts were prepared with an NP40 lysis buffer solution (Boston BioProducts, Milford, MA, USA). For the determination of the protein concentration, the DC protein assay kit (Bio-Rad, Munich, Germany) was used. Protein lysates were separated according to their size by SDS-PAGE in 4–20% precasted gels (Thermo Fisher Scientific) using a voltage between 120 V and 175 V. After electrotransfer onto nitrocellulose membranes (Amersham/GE Healthcare, Amersham, UK) at 65 V for two hours, the membranes were stained in the Ponceau staining solution until protein bands were visible, and then cut to useful sizes and pieces depending on the proteins in focus. After the addition of the respective primary antibody in a dilution of 1:1000, the membranes were incubated over night at 8 °C. Secondary antibodies were added at a dilution of 1:5000 and incubated at room temperature for one hour. After washing, the membranes were developed using the LAS 4000 imager (GE Healthcare Bio-Sciences, Pittsburgh, PA, USA) and Super Signal West Dura chemiluminescence substrate kit (Thermo Fisher Scientific).

#### 2.2.7. Lipidomic Chromatography

The fatty acid composition of cell extracts was determined by gas chromatography coupled with mass spectrometry. Cell pellets were resuspended in 50 µL of water, and after adding 100 µL internal standard mix (tetradecanoate d27 and heptadecanoate d33, 200 µg/mL each in Methanol/Toluol 4/1) as well as 1000 µL Methanol/Toluol 4/1, the cells were vortexed. Then, 100 μL acetyl chloride was added and the samples were mixed vigorously. Subsequently, the samples were heated for 1 h at 100 °C to prepare fatty acid methyl esters. After cooling to room temperature, 3 mL of 6% sodium carbonate was added and the samples were mixed vigorously. The mixture was centrifuged (1800× *g*, 5 min) and the upper layer was transferred to auto sampler vials. Gas chromatography analyses were performed using a Trace 1310 gas chromatograph (Thermo Fisher) equipped with the following stationary phase: DB-225 30 m × 0.25 mm i.d., film thickness 0.25 µm (Agilent) coupled to a mass spectrometer (ISQ 7000 GC-MS, Thermo Fisher Scientific, Dreieich, Germany). Peak identification and quantification were performed by comparing retention times and peak areas, respectively, to standard chromatograms and internal standards.

### 2.3. Mass Spectrometry-Based Differential Quantitative Proteome Analysis

#### 2.3.1. Protein Extraction and Tryptic Digestion

The Kasumi1 cell line samples were dissolved in 100 mM triethyl ammonium bicarbonate and 1% *w*/*v* sodium deoxycholate buffer, boiled at 95 °C for 5 min and sonicated with a probe sonicator. The protein concentration of denatured proteins was determined by the Pierce BCA Protein assay kit (Thermo Fisher). Then, 20 μg of protein for each sample was diluted in 50 μL buffer containing 0.1 M TEAB and 1% (*w*/*v*) SDC in H_2_O. Disulfide bonds were reduced with 10 mM DTT at 60 °C for 30 min. Cysteine residues were alkylated with 20 mM iodoacetamide (IAA) for 30 min at 37 °C in the dark. Tryptic digestion was performed for 16 h at 37 °C, using a trypsin/protein ratio of 1:100. After tryptic digestion, the inhibition of trypsin activity as well as the precipitation of SDC was achieved by the addition of 1% formic acid (FA). Samples were centrifuged for 5 min at 16,000× *g*. The supernatant was dried in a SpeedVac vacuum concentrator and stored at −20 °C until further use. Prior to mass spectrometric analyses, peptides were resuspended in 0.1% FA to a final concentration of 1 µg/µL. Finally, 1 µg was used for LC-MS/MS acquisition.

#### 2.3.2. LC-MS/MS Acquisition and Data Processing

The chromatographic separation of peptides was achieved by nano-UHPLC (Dionex Ultimate 3000 UHPLC system, Thermo Fisher) with a two-buffer system (buffer A: 0.1% FA in water, buffer B: 0.1% FA in ACN). Attached to the UPLC was a peptide trap (100 µm × 20 mm, 100 Å pore size, 5 µm particle size, Acclaim PepMap, Thermo Scientific) for online desalting and purification, followed by a 25 cm C18 reverse-phase column (75 µm × 200 mm, 130 Å pore size, 1.7 µm particle size, Peptide BEH C18, Waters). Peptides were separated using an 80 min method with a 60 min gradient elution from 2% to 30% buffer B. The eluting peptides were analyzed on a Quadrupole Orbitrap hybrid mass spectrometer (QExactive, Thermo Fisher Scientific). Here, the ions being responsible for the 15 highest signal intensities per precursor scan (1 × 10^6^ ions, 70,000 Resolution, 240 ms fill time) within a scan range from 400 to 1200 *m*/*z* were analyzed by MS/MS (HCD at 25 normalized collision energy, 1 × 10^5^ ions, 17,500 Resolution, 50 ms fill time) starting at 120 *m*/*z.* A dynamic precursor exclusion of 20 s was used.

LC-MS/MS data were searched with the Sequest algorithm integrated in the Proteome Discoverer software (Version 3.0.0.757, Thermo Fisher Scientific) against a reviewed human Swissprot database, obtained in December 2022. Carbamidomethylation was set as fixed modification for cysteine residues and the oxidation of methionine, and pyro-glutamate formation at glutamine residues at the peptide N-terminus, as well as acetylation of the protein N-terminus were allowed as variable modifications. A maximum number of 2 missing tryptic cleavages was set. Peptides between 6 and 144 amino acids were considered. A strict cutoff (FDR < 0.01) was set for peptide and protein identification. Quantification was performed using the Minora Algorithm, implemented in the Proteome Discoverer Software.

The mass spectrometry proteomics data were deposited to the ProteomeXchange Consortium via the PRIDE partner repository with the dataset identifier PXD048252 [23].

#### 2.3.3. Kinome Analysis

Functional kinome profiling of tyrosine as well as serin–threonin kinases has been described previously [24]. Here, we used a PamStation^®^12 (located at the UCCH Kinomics Core Facility, Hamburg, Germany) and PTK-PamChip^®^ arrays to profile tyrosine kinase according to the manufacturer’s instructions (PamGene International, s-Hertogenbosch, The Netherlands). In brief, whole-cell lysates were made using 100 μL M-PER Mammalian Extraction Buffer containing Halt phosphatase inhibitor and ethylenediaminetetraacetic acid (EDTA)-free Halt protease inhibitor cocktail (1:100 each; Pierce, Waltham, MA, USA) per 1 × 10^6^ cells. The lysed sample were stored immediately in a −80 °C freezer. Protein quantification was performed with the bicinchoninic acid assay according to the manufacturer’s instructions (BCA; Merck KGaA, Darmstadt, Germany). Per array 5 μg of protein and 400 μM ATP were applied. Sequence-specific peptide tyrosine and serin–threonin phosphorylation were detected by the fluorescein-labeled antibody PY20 (Exalpha, Maynard, MA, USA) and a CCD camera using the Evolve software (1200PTKlysv04.PS12Protocol; PamGene International, s-Hertogenbosch, The Netherlands). Data were analyzed using the BioNavigator software (PamGene International, s-Hertogenbosch, The Netherlands).

### 2.4. Statistical Analysis

For evaluating significance, an unpaired t-test was performed. Significance is presented in the graphs. Standard deviation is presented as error bars. The ratio between p-protein and total protein was calculated in order to show the authentic changes of phosphorylation which is not due the differences in total protein that can be induced by protein expression. In order to better standardize the Western blot quantification, the analysis of data is based on the total protein quantification using Ponceau red staining.

#### 2.4.1. Statistical Analysis of Proteomic Data

Normalized protein abundances were analyzed within the statistic software Perseus (Version 2.0.11.0). Abundances were log2-transformed and reduced to only valid values to perform linear principal component analysis. For statistical testing, data were reduced to proteins found in more than 2 replicates per phenotype (scr, kd1, kd2). Student’s *t*-testing, including permutation-based FDR correction, was performed. As the threshold for *p*- and adjusted *p*-values, 0.05 was applied as well as a 2-fold change. Visualizations were performed in R (Version 4.3.3) environment [25] using an in-house script based on the ggplot^2^ package. For the analysis of FASN regulation, visualization was performed in GraphPad Prism (Version 8.0.2) as well as *t*-testing, with significances of ** for *p* < 0.01 and *** for *p* < 0.001.

#### 2.4.2. Software

AIDA Image Analyzer: Version 3.44.

Zotero: 6.0.20.

## 3. Results

### 3.1. Establishment of Stable FASN Knockdown in Kasumi1 by Proteomic Analysis

The major goal of this project was the analysis of the functional role of FASN for signal transduction in CBF-AML cells. Therefore, the stable knockdowns of FASN were established in the AML cell line Kasumi, an AML cell line with t(8;21) and c-Kit mutation at Asn822. As shown in Figure 1, the expression of FASN was reduced by two independent lentiviral knockdown vectors by over 80%.

The biological effect of FASN knockdown on the levels of fatty acids was analyzed in Kasumi1 cells through chromatography experiments. Twenty-seven fatty acids were detected, and their amounts (in µg/µL) were normalized to the total concentration of cellular fatty acids and quantified. In Figure 2, the relative amount of myristic and palmitic acids is shown in Kasumi1 cells after the knockdown of FASN (kd1 and kd2) compared to Kasumi1 cells without knockdown (scr). A relative reduction in fatty acid levels from 100% to 64% and 86% for myristic acid, and from 100% to 85% and 85% for palmitic acid, respectively, was shown, confirming the successful knockdown of FASN in both cell clones. The other fatty acids were not downregulated by FASN knockdown.

We also reproduced FASN knockdowns by mass spectrometry in both kd1 and kd2 cell clones (Figure 3).

### 3.2. Upregulation of p-c-Kit, pLyn, pMAPK, and pS6 in Kasumi1, and in Baf3 c-Kit N822K Cells After Pharmacological and Genetic FASN Inhibition

Using Western blot, a significant upregulation of phosphor-S6 (pS6) was observed in both Kasumi1 FASN kd1 and kd2 cells. On the contrary, phospho-Akt (pAkt) and phospho-MAPK (pMAPK) did not show any significant changes (Figure 4A). We also observed a significant upregulation of pS6 after treatment with the FASN inhibitor TVB-3166; however, the effect occurred 8 h after treatment with TVB-3166 but not after 48 h (Figure 4B). As mentioned above, we also investigated whether there is a link between FASN activity and the Hedgehog pathway, exemplified here by determining the expression of Gli1 in FASN-knockdown cells. As shown in Figure 4C, we did not see any significant changes in Gli1 expression levels under the two stable FASN-knockdowns in Kasumi1 cells.

To verify our results, we examined the aberrant signaling of mutant Kit in two separate sub-cell lines of Baf3 cells. Both cell lines ectopically expressed Kit-N822K but were generated independently of each other (Figure 5A,B). In the comparable design, the pharmacological inhibition of FASN by TVB-3166 treatment resulted in an upregulation of pS6 (Figure 5A,C). However, the agonistic effect of FASN inhibition occurred much later in the Baf3-N822K cells compared to the Kasumi1 cells (Figure 4), suggesting that the time course of pharmacological FASN inhibition is different in the different cellular models and consequently the downstream signaling of the Kit mutant might also be affected, at least temporally differently.

In further experiments, the tyrosine as well as serin–threonin kinases regulated by FASN in Kasumi1 cells were identified in detail by functional kinome profiling (Figure 6 and Figure 7). The expression of FASN was either reduced by knockdown, or the activity of FASN was inhibited by TVB-3166. As shown in Figure 6A,B and Figure 7A,B, respectively, kinase activity was increased both after sustained knockdown, as well as after treatment with 100 nM FASN-inhibitor TVB-3166 for 5 days. However, it must be noted that only kd2 cells could be analyzed further for their regulatory effects on serin/threonin kinases, since basically no upregulation was detected in kd1 cells and only weak upregulation was detected in TVB-3166-treated cells (Figure 7B). Notably, kinome profile analysis revealed an increased activity of p-Kit, several members of the SRC family kinases (SFKs, including pLyn), and pS6, along with upregulation of pMAPK, both in Kasumi1 FASN knockdowns and following treatment with TVB-3166. Additionally, a marked upregulation of protein kinase A (PKA), a key component of cAMP-dependent signaling, was observed (Figure 6C and Figure 7C).

### 3.3. No Changes in the Proliferation Activity of Kasumi1 Cells After the Inhibition of FASN

In the next experiments, the effect of FASN on the proliferation of Kasumi1 cells was analyzed.

Despite the repeatedly observed effect on the downstream signaling of KIT N822K, as well as the activation status of other kinases, we did not observe significant changes in the proliferation of Kasumi1 cells after FASN knockdown or after treatment with 100 nM TVB-3166 (Appendix A by “Appendix A”).

### 3.4. Reduction in Kasumi1 Cell Viability by Combination of FASN Inhibitor TVB-3166 with Gli1 Inhibitor GANT61

In subsequent experiments, we examined the effects of FASN and Gli1 inhibition on the viability of Kasumi1 cells. As single agents, TVB-3166 (at concentrations of 10 nM and 25 nM) and GANT61 (at concentrations of 5 μM and 10 μM) did not significantly impact cell viability. However, when combined, these compounds led to a marked significant decrease in cell viability, even at these low concentrations (Figure 8).

## 4. Discussion

Activating c-Kit mutations predominantly occur in CBF-AML, causing an adverse clinical outcome [26,27]. To address the limitations in treating CBF-AML patients with additional KIT mutations, particularly those in the kinase activation loop (exon 17), further investigation into additional treatment options is essential.

As mentioned above, active c-Kit mutants are usually localized on the Golgi network leading to an activation of the main signaling pathways MAPK, Akt, and STAT5 [11]. Lyn, a non-receptor tyrosine kinase (Non-RTK) and member of the Src kinase family, plays a crucial role as a co-effector of c-Kit. It participates in c-Kit receptor-mediated signaling by trafficking through the Golgi and associating with the juxtamembrane region of c-Kit, a process regulated by the SH4 domain of Lyn [28]. Lyn was found to be highly expressed in a large cohort of AML patients, playing a crucial role in leukemogenesis. It has been reported to localize throughout the plasma membrane and cytoplasm of AML cells, where it is expressed in an active state [29]. The inhibition of Lyn markedly decreased the phosphorylation of mTOR and STAT5, while leaving Akt phosphorylation unaffected. Furthermore, Lyn is a key signaling component in FLT3-ITD-mutated AML, particularly within the FLT3-ITD-STAT5 pathway [29,30].

FASN-mediated palmitoylation of target proteins is recognized as a significant mechanism underlying resistance, particularly in TKI-resistant, epidermal growth factor receptor (EGFR)-mutated non-small cell lung cancer. The combination of EGFR-TKIs with the FASN inhibitor Orlistat effectively disrupted EGFR-TKI resistance in vivo [31]. Meanwhile, diverse FASN inhibitors were developed for clinical studies [32,33,34]. Intriguingly, the interrelationship between S-palmitoylation and cell signaling transduction, as well as its biological role are poorly understood in hematological malignancies. In mice with an AML phenotype, it has been reported that the S-palmitoylation of the GTPases H-Ras and N-Ras, or of the mutated receptor FLT3-ITD is essential for their binding, localization, and trafficking between the plasma membrane and different endosomes [35,36]. In 2019, it has been reported that in AML with an activating c-Kit N822K mutation, the mutant is active on the Golgi, leading to an activation of Akt, MAPK, and STAT5 signaling, while the inactive/dephosphorylated c-Kit form is localized in the ER [37]. However, the role of palmitoylation for the localization of the c-Kit mutant on the Golgi lipid rafts and for its subsequent activity is currently under investigation.

We established two stable FASN knockdowns in Kasumi1 cells through lentiviral transduction with FASN-shRNA vectors. Their statistical and biological significance was confirmed by immunoblotting, mass spectrometry, and lipidomic analysis, which showed a numeric reduction in myristic and palmitic acid levels in both FASN knockdowns. Using two cell models of c-Kit-N822K-mutated CBF-AML, we demonstrated that both pharmacological and genetic FASN inhibition resulted in the rewiring of c-Kit-associated pathways, particularly with a significant upregulation of p-c-Kit, pLyn, pMAPK, and pS6. Recent studies suggest a potential role for S-palmitoylation in lysosomal functions, primarily through its effects on proteins involved in lysosomal trafficking, membrane fusion, signaling pathways, and protein degradation [38]. The inhibition of FASN may attenuate the degradation of c-Kit and p-c-Kit via this mechanism. The subsequent upregulation of pMAPK and pS6 should be interpreted as an mTOR-independent pathway driven by the pronounced upregulation of PKA, as demonstrated by our kinomic analysis. The cAMP-PKA pathway is known to typically suppress mTOR activity, as reported in several studies [39]. More likely, the upregulation of pS6 is mediated by increased pMAPK following FASN knockdown, as supported by published evidence linking S6 kinase activation to MEK/ERK signaling in malignant cells [40]. Our findings suggest that the rewiring of c-Kit-associated signaling upon FASN inhibition leads to a heightened dependence of AML cells on pLyn-pS6 signaling. Consistent with these observations, no significant changes in Kasumi-1 cell proliferation were observed following treatment with the FASN inhibitor TVB-3166 or FASN knockdown. This lack of response may be additionally attributed to the well-demonstrated anti-apoptotic function of pS6, which phosphorylates BAD (Bcl-2-associated agonist of cell death) in AML [41].

Given the clinical approval of the Hedgehog signaling inhibitor Glasdegib for AML treatment, we explored whether the expression of the downstream Hedgehog effector Gli1 is dependent on FASN activity to identify potential additional treatment strategies. Our hypothesis is supported by reports from studies which have shown that FASN knockdown via siRNA can reduce Gli1 levels in gastric cancer cells, suggesting that FASN may contribute to tumorigenesis and metastasis [42]. However, we did not observe any reduction in Gli1 expression following FASN inhibition. This may be attributed to a compensatory effect of increased pS6, supporting the existence of SMO-independent, non-canonical regulation of Hedgehog-Gli signaling by key oncogenic drivers, as demonstrated in various myeloid malignancies [43]. To further explore the hypothesis that FASN inhibition increases dependence on the pLyn-pS6-Gli1 pathway for cell survival, we evaluated the effects of TVB-3166 and the Gli1 inhibitor GANT61 on the viability of Kasumi1 cells, both individually and in combination. While neither compound alone has significantly impacted Kasumi1 cell viability, their combination resulted in a substantial reduction. Given that both agents were administered together at low concentrations, we suggest that this cytotoxic effect is specific and biologically significant.

## 5. Conclusions

In this study, we examined the role of fatty acid synthase (FASN) in c-Kit signaling in both human and murine acute myeloid leukemia (AML) cell lines with activating c-Kit mutations. The inhibition of FASN, either through the compound TVB-3166 or shRNA, led to a notable increase in the phosphorylation of Lyn (pLyn), a member of the Src kinase family, alongside elevated levels of p-c-Kit, pMAPK, and pS6. Consistent with previous reports, Gli is upregulated by S6 kinase in a non-canonical manner [43]. In line with these observations, we found that FASN inhibition resulted in sustained Gli1 expression and continued proliferation of Kasumi1 cells, likely driven by the upregulation of the pLyn-pS6 signaling axis. Additionally, combining TVB-3166 with the Gli1 and Gli2 inhibitor GANT61 markedly reduced Kasumi1 cell survival. To our knowledge, this is the first report of the upregulation of p-c-Kit, Src/pLyn, pMAPK, and pS6 in the context of FASN inhibition, alongside the distinct biological behavior observed in Kasumi1 cells. Further research is required to fully elucidate how FASN inhibition reprograms c-Kit-associated signaling pathways.

## Figures and Tables

**Figure 1 biomedicines-13-00619-f001:**
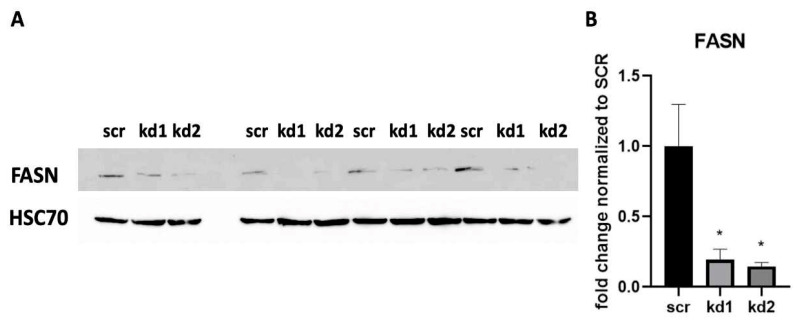
shRNA-mediated knockdown of endogenous FASN in Kasumi1 cells. FASN expression was detected in whole cell lysate by Western blot. Kasumi1 FASN kd cell lines were transduced with FASN shRNA (either with vector1 which is kd1 or with vector2 which is kd2) or Kasumi1 scr cell line transduced with non-targeting vector as a control for the baseline expression of FASN in the Kasumi1 cell line. HSC70 was used as the loading control for the quantification of FASN expression. The Western blot is shown in (**A**) and the analysis in (**B**), with y-axis of fold change normalized to the expression level of FASN in scr. Each sample was loaded three times and analyzed by a *t*-test. Significance is presented in the graphs as * for *p* < 0.05. Ponceau staining of the same membrane was used to demonstrate the equal loading of protein lysates (see “Appendix A”).

**Figure 2 biomedicines-13-00619-f002:**
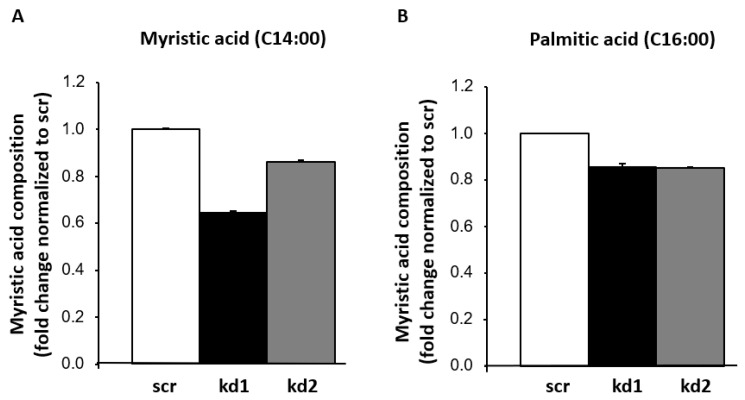
Lipidomic analysis in Kasumi1 cells after the knockdown of FASN. Concentrations of myristic (**A**) and palmitic acids (**B**) were detected in relation to the total amount of fatty acids and normalized to scr. All samples were analyzed in technical duplicates. A relative reduction in both fatty acids by FASN-knockdown kd1 and kd2 vs. scr could be detected.

**Figure 3 biomedicines-13-00619-f003:**
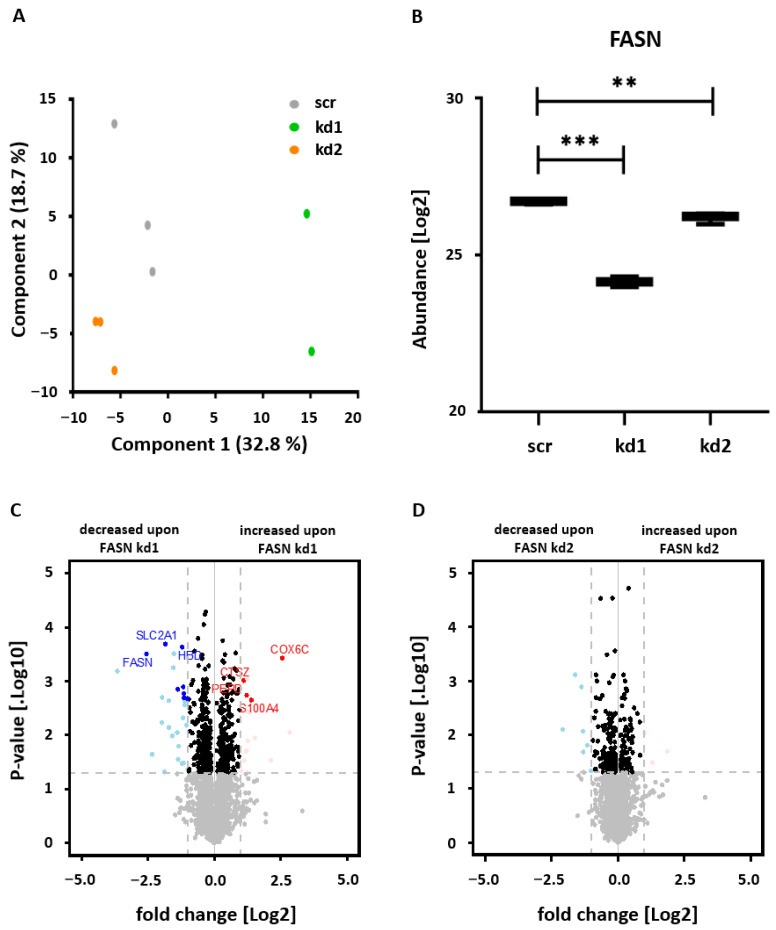
Proteome analysis of Kasumi1 cells after FASN knockdown. (A) Scatter plot visualization of principal component analysis (PCA), showing components 1 and 2. The plot reveals the phenotype-based clustering of scr, kd1, and kd2 groups. The analysis was performed using 2155 valid quantified proteins. (B) Log2 scale representation of FASN abundance in scr, kd1, and kd2. (C) Volcano plot visualization of Student’s *t*-test results between kd1 and scr, showing significant downregulation of FASN. (D) Volcano plot comparing kd2 and scr, based on 2314 proteins, also showing significant downregulation of FASN. Proteins were considered significantly differentially abundant if they exceeded the adjusted *p*-value cutoff of <0.05 and >2-fold change. Significance was assessed using an unpaired *t*-test as ** *p* < 0.01, *** *p* < 0.001.

**Figure 4 biomedicines-13-00619-f004:**
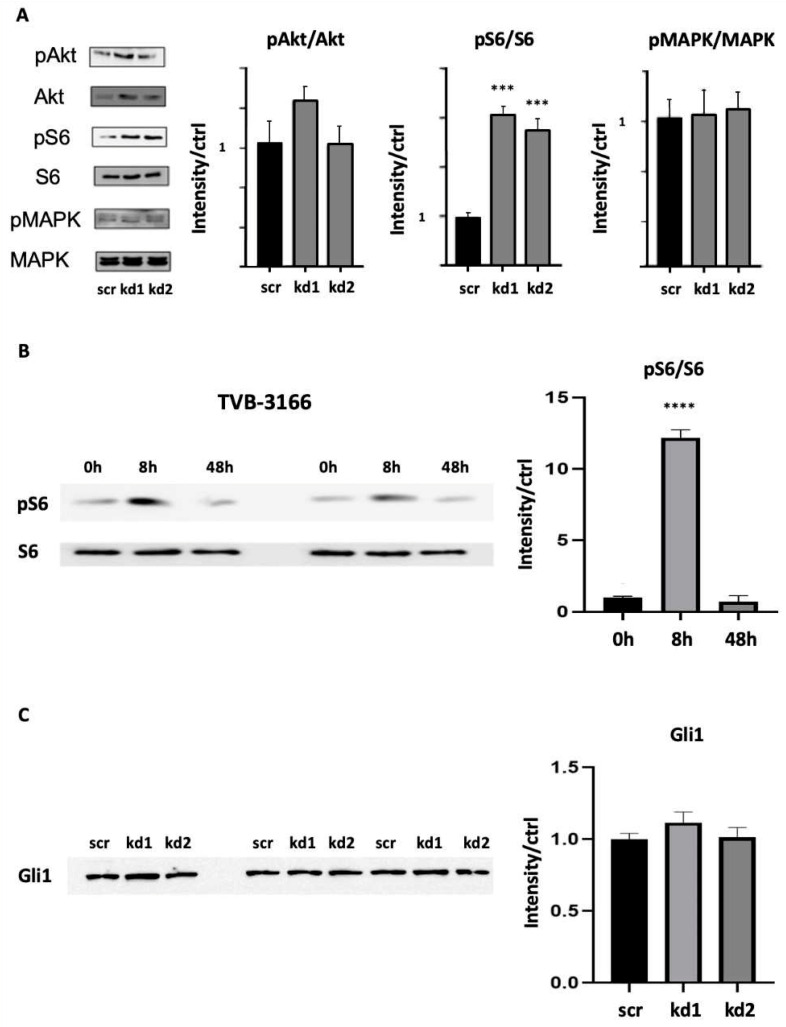
Analysis of c-Kit signal transduction in Kasumi1 cells by Western blotting. (**A**) Increased phosphorylation of S6 after the stable knockdown of FASN in Kasumi1 cells. No significant regulation of Akt and MAPK was detected in Kasumi1 cells with stable FASN-knockdown kd1 and kd2, and scr control cells. The samples were analyzed in technical triplicates. One of each of the three triplicates are shown on the left, and the statistic results are shown on the right. Densitometric quantification of phospho-protein/protein ratios after normalization to scr control cells (n = 3, mean values with standard deviations). Significance is presented as *** for *p* < 0.001. (**B**) Increased phosphorylation of S6 after treatment of Kasumi1 cells with 100 nM TVB-3166. Protein lysates were analyzed for the expression of pS6 and total S6 (S6) protein in technical replicates. Statistic bar charts used the optical density normalized to time 0 h. Significance is presented in the graphs as **** *p* < 0.001. (**C**) No significant regulation of Gli1 in Kasumi1 cells with stable FASN knockdown was analyzed in technical triplicates. Densitometric quantification of Gli1 expression relative to scr control cells (n = 3, mean values with standard deviation). Ponceau staining of the shown membranes was used to demonstrate the equal loading of protein lysates (see “Appendix A”).

**Figure 5 biomedicines-13-00619-f005:**
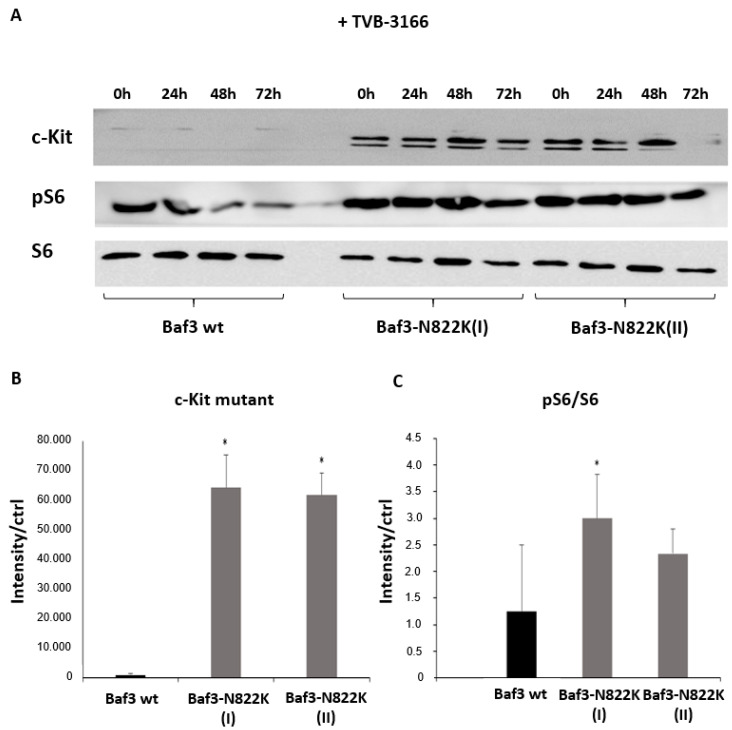
Analysis of c-Kit N822K-mediated signal transduction in Baf3 cells by Western blotting. Protein lysates were analyzed for the expression of c-Kit and phosphorylation of Akt and S6 in Baf3 wild type (Baf3 wt), as well as in two Baf3 cell cultures independently transduced with the c-Kit mutant N822K (I and II). (**A**) Expression of the N822K mutant, and phosphorylation of Akt and S6 were explored by using phospho-specific antibodies. Significantly higher levels of N822K mutant expression (**B**) as well as phosphorylated S6 (**C**) after treatment with 100 nM TVB-3166 in Baf3 cells with c-Kit gain-of-function mutation N822K in comparison to Baf3 wt were detected. Ponceau staining of the shown membranes was used to demonstrate equal loading of protein lysates (see “Appendix A”). Protein lysates were analyzed in technical triplicates determined as a mean value with the standard deviation of the time points 0, 24, and 48 h in each of the three approaches Baf3 wt, Baf3-N822K I and II, respectively. Significance is presented in the graphs as * = *p* < 0.01.

**Figure 6 biomedicines-13-00619-f006:**
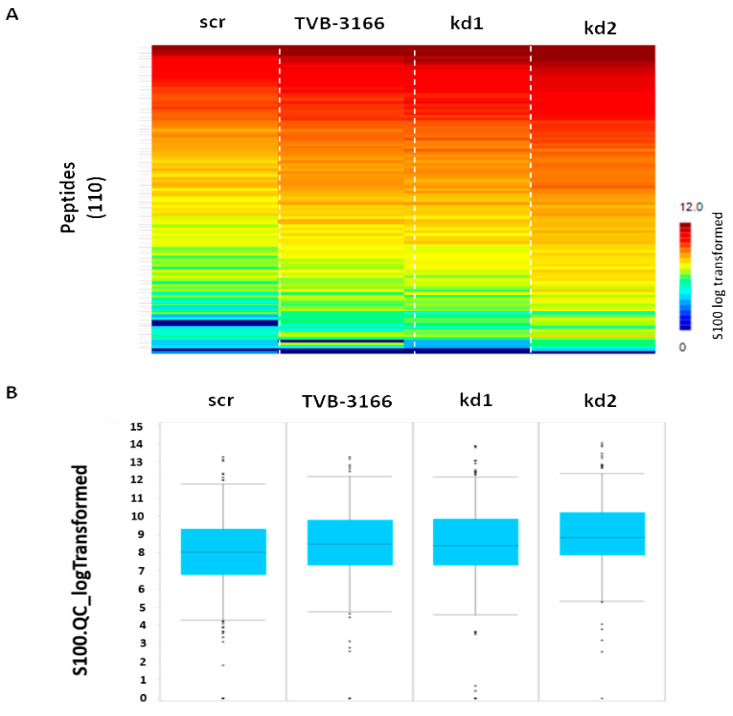
Kinome analysis of tyrosine kinases in Kasumi1 cells. Tyrosine kinases were analyzed in Kasumi1 cells with FASN knockdowns (kd1 and kd2) or after treatment with 100 nM TVB-3166 using functional kinome profiling. (**A**) The heatmap of 110 analyzed peptides (S100_log-transformed values are depicted). (**B**) Boxplot quantifying the mean signal intensities. (**C**) Upstream kinase analysis of control vs. TVB-3166, FASN knockdown 1 or 2, respectively (normalized kinase statistic (log2) > 0: higher kinase activity in treated samples; specificity score (log2) > 1.3 (white-to-red circles): statistically significant changes).

**Figure 7 biomedicines-13-00619-f007:**
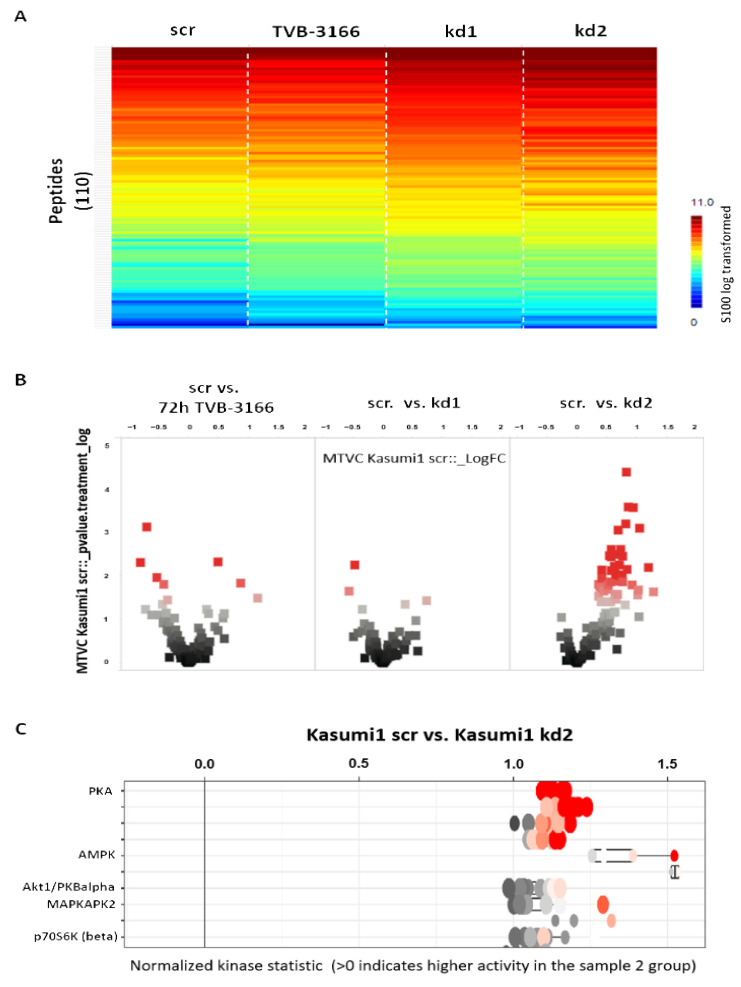
Kinome analysis of serin/threonin kinases in Kasumi1 cells. Serin/threonin kinases were analyzed in Kasumi1 cells with FASN knockdowns (kd1 and kd2) or after treatment with 100 nM TVB-3166 using functional kinome profiling. (**A**) The heatmap of 110 analyzed peptides (the S100_log transformed values are depicted). (**B**) Volcano plot highlighting significantly altered peptides (scr control vs. treatment; x-axis: log fold change in peptide phosphorylation, dashed line = 0; *y*-axis: significance (plog) for each peptide, >1.3 (dashed/dotted line) significant changes). (**C**) Upstream kinase analysis of scr vs. kd2 (normalized kinase statistic (log2) < 0: lower kinase activity in inhibitor treated sample; specificity score (log2) > 1.3 (white to red circles): statistically significant changes).

**Figure 8 biomedicines-13-00619-f008:**
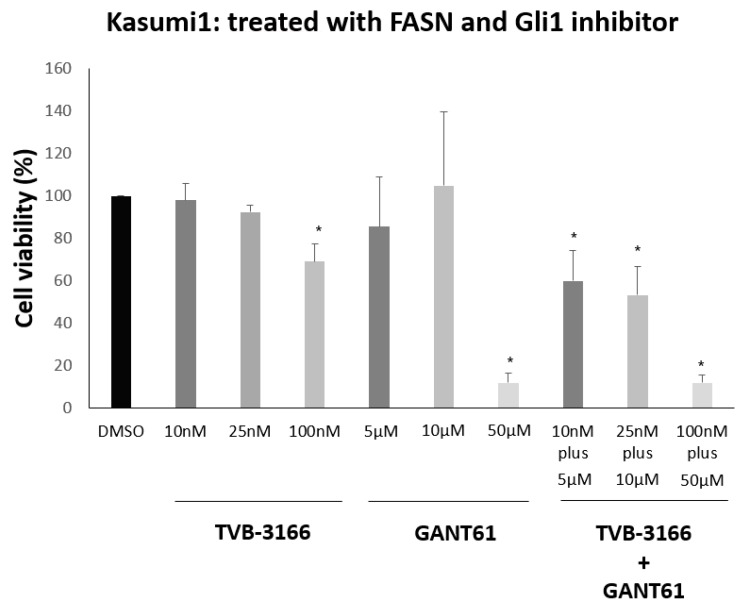
Kasumi1 cell viability upon FASN inhibitor TVB-3166 and Gli1 inhibitor GANT61. Cell viability was assessed on day 5 following treatment with inhibitors using the Vi-CELL™ XR via a trypan blue exclusion assay. Single-agent treatments with TVB-3166 (10 nM, 25 nM) and GANT61 (5 μM, 10 μM) showed no significant impact on cell viability. However, the combination of both compounds led to a significant reduction in viability compared to TVB-3166 alone across all concentrations. The results represent biological triplicates, with the mean values ± standard deviation. Statistical significance is indicated in the graphs by * for *p* ≤ 0.05.

## Data Availability

The original contributions presented in this study are included in the article/Appendix A. Further inquiries can be directed to the corresponding author.

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
