# Peer review of "Functional Role of Fatty Acid Synthase for Signal Transduction in Core-Binding Factor Acute Myeloid Leukemia with an Activating c-Kit Mutation"

_biomedicines, 2025, doi:10.3390/biomedicines13030619_

Round 1
Reviewer 1 Report
Comments and Suggestions for Authors
In this study, the role of fatty acid synthase (FASN) in two c-Kit-N822K-mutated acute myeloid leukemia (AML) cell lines was investigated using FASN knockdown via shRNA and the FASN inhibitor TVB-3166. Functional implications, including cell proliferation, were assessed through western blotting, mass spectrometry, and PamGene. FASN inhibition led to increased phosphorylation of c-Kit (p-c-Kit), Lyn kinase (pLyn), MAP kinase (pMAPK), and S6 kinase (pS6). Furthermore, sustained high expression of Gli1 in Kasumi1 cells following FASN inhibition was observed, which is well known to be mediated by the upregula-tion of pS6. This study is interesting and rich in content. Please modify the manuscript carefully according to the requirement of the journal. Followings are some suggestions for revisions.
- In the last paragraph of introduction section, the key innovations of present work can be emphasized, and it is better not to describe the results of present work in this part.
- The “Methods” can be changed to “Materials and Methods”. And under this section, a subsection entitled “Materials and reagent” can be used to cover “Bacterial strain”, “Kits”, “Antibodies”, and “Vectors”, these subtitles can be removed and the content can be combined together.
- There should be a space between the numeral and unit, including in the text and figures. The “μl”, “ml”, and “minutes” can be changed to “μL”, “mL”, and “min”, respectively. The “m/z” should be in Italics font.
- The resolution and quality of figures should be further improved. For example, the quality of Figure 3 should be improved. In Figure 5, the description in the graphics can be provided in the figure legend, and mark the graphics ABC.
- The conclusion section can be improved. More content can be added.
- The format of the manuscript should be revised according to the requirement of the journal. It is suggested to use the template provided by the journal.
- More experimental data can be provided as tables, at least as supplementary material, such as the GC-MS and LC-MS analysis results. The typical chromatograms of GC-MS and LC-MS analysis can also be provided in the supplementary material.
Author Response
Dear Reviewer 1,
We would like to sincerely thank you for your review of our manuscript. Please find below our point-by-point response to your comments. We hope that the revisions we have made address your concerns satisfactorily in this revised version.
Best regards,
Maxim Kebenko

Reviewer 2 Report
Comments and Suggestions for Authors
This manuscript shows the relevance of the fatty acid synthase (FASN) on the c-kit associated pathways in a leukemia cell line (kasumi1). The study shows that inhibition of FASN does not interfere with cell proliferation, but phosphorylation of diverse proteins associated with c-kit signaling pathways was upregulated (p-c-Kit, pLyn, pMAPK, and pS6). Nonetheless, combination of TVB-3166 and GANT61 inhibitors for FASN and Gli1 significantly reduced cell viability at rather low to moderate concentrations.
The results of this study are interesting and open the way for additional concerns on this issue to be investigated. The experimental set-up is appropriate and well described in the manuscript.
A few emendments need to be accomplished before the manuscript is accepted for publication as follows:
1) Figures 6 and 7 need quality improvement.
2) Page 11: revise the text "We also observed a significant upregulation of pS6 was after treatment with the FASN inhibitor TVB-3166".
3) Figure 8: it seems that the effect of both inhibitors, TVB-3166 and GANT61, at 100nM and 50uM is equivalent to that of only GANT61 and, consquently, the effect of TVB-3166 seems to be disguised or negligible. The effect of GANT61 at lower concentration in the combination with TVB-3166 at 100 nM would be of interest.
Author Response
Dear Reviewer 2,
We would like to sincerely thank you for your review of our manuscript. Please find below our point-by-point response to your comments. We hope that the revisions we have made address your concerns satisfactorily in this revised version.
Best regards,
Maxim Kebenko

Reviewer 3 Report
Comments and Suggestions for Authors
The paper is interesting and the data compelling. However, quality of figures is poor, as well as the disposal of panels. Following my observations.
Please explain FASN abbreviation also in background.
Please name (e.g. A, B..) the panels in the figure.
Quality of figure 3 is poor. Furthermore, PCA should be further commented in both legend and text. Furthermore, panels C and D could benefit from a clearer statement of the groups compared.
Figure 4 is after figure 5, 7 is after 8.
Quality of figures 6 and 7 is very poor and, especially panel C is difficult to read. Furthermore, the panels should be shown in the order of the reference in the text.
Supplementary materials have no legends or captions for the figure.
Author Response
Dear Reviewer 3,
We would like to sincerely thank you for your review of our manuscript. Please find below our point-by-point response to your comments. We hope that the revisions we have made address your concerns satisfactorily in this revised version.
Best regards,
Maxim Kebenko
